# The Influence of Cultural Experiences on the Associations between Socio-Economic Status and Motor Performance as Well as Body Fat Percentage of Grade One Learners in Cape Town, South Africa

**DOI:** 10.3390/ijerph19010121

**Published:** 2021-12-23

**Authors:** Eileen Africa, Odelia Van Stryp, Martin Musálek

**Affiliations:** 1Faculty of Medicine and Health Sciences, Stellenbosch University, Private Bag X1, Matieland 7602, South Africa; africa@sun.ac.za (E.A.); odeliavs@sun.ac.za (O.V.S.); 2Faculty of Physical Education and Sport, Charles University, José Martího 31 Praha 6, 162 52 Veleslavín, Czech Republic

**Keywords:** fundamental movement skills, physical fitness, adiposity, children, cultural experiences, socio-economic status

## Abstract

Fundamental movement skills (FMS), physical fitness (PF) and body fat percentage (BF%) are significantly related to socio-economic status (SES). However, it remains unclear why previous studies have had different findings regarding the direction of the association between SES and FMS, PF and BF%. A suggested explanation is that the direction of the link can be influenced by cultural experiences and traditions. Therefore, the aim of the current study was to investigate links between SES and FMS, PF, BF% of Grade One learners from two different ethno-geographic areas in Cape Town, South Africa. Grade One children (*n* = 191) (*n* = 106 boys and *n* = 85 girls; age (6.7 ± 0.33)) from different socio-economic areas in Cape Town, South Africa, were selected to participate in the study. South African schools are classified into five different quintiles (1 = poorest and 5 = least poor public schools). For this study, two schools were selected, one from quintile 2 and the other from quintile 5. BF% was assessed according to Slaughter’s equation. FMS were measured using the Gross Motor Development Test-2 (TGMD-2) and PF via five tests: 1. dynamic strength of lower limb (broad jump); 2. dynamic strength of upper limb and trunk (throwing a tennis ball); 3. speed agility (4 × 10 m shuttle running); 4. cardiorespiratory fitness (20 m shuttle run endurance test (Leger test)) and 5. flexibility (sit and reach test). An analysis of covariance (ANCOVA) found that BF% and WHtR were significantly greater in children with higher SES (Z = 6.04 *p* < 0.001; Hedg = 0.54), (Z = 3.89 *p* < 0.001; Hedg = 0.32). Children with lower SES achieved significantly better TGMD-2 standard scores in the locomotor subtest, compared to their peers with higher SES. In the object control subtest, no significant SES-related difference was found. However, ANCOVA showed that girls performed better in FMS than boys. In PF, the main effect of SES was observed in dynamic strength of trunk and upper limb (throwing) and flexibility, where children with lower SES performed significantly better. No significant difference was found in cardiorespiratory performance (CRP) (Beep test), even though children with lower SES achieved better results. Results from the current study suggest that links between SES, PF, FMS and body fat percentage in children seem to be dependent on cultural and traditional experiences. These experiences should therefore be included as an important factor for the development of programmes and interventions to enhance children’s lifelong motor behaviour and health strategies.

## 1. Introduction

The development of fundamental movement skills (FMS) and health-related physical fitness (PF) during childhood presents important health parameters [1] for promoting long-term positive and sustainable health trajectories, especially in obesity prevention [2]. In addition, a large body of studies have verified that children with lower PF or FMS, regardless of geographic specification, tend to be overweight or obese [3,4,5,6,7], even at pre-school age [8,9]. The aforementioned health trajectories are expounded in the Stodden model [10], which explains that reciprocal relations between motor competence, FMS, health-related physical fitness, self-perceiving of motor competences and physical activity play a key role. However, the Stodden model does not consider the socioeconomic status (SES) of individuals, although SES has been found to be an important indicator in the prevalence of obesity as well as in PF and FMS development [11].

Previous research that focused on the relationship between obesity and SES in school-age children has not brought clear results. While studies that include children from North America, Australia and Europe suggest that lower SES is significantly associated with a risk of overweightness and obesity [9,12,13], other studies from Brazil or Korea did not find any such relationship [14,15]. Moreover, research done in South America [16], the Arab world [17], and particularly in Africa [11,18,19] showed reversed patterns. It means that children with high SES displayed a higher chance of being overweight and obese. One of the suggested explanations for this research disconformity is that obesity is strongly associated with globalization, i.e., expansion of economic and social interdependence [19,20], which is generally not the same in different world regions.

When we look at the association between motor performance and SES, one of the basic assumptions is that children with lower SES tend to have motor developmental delays [21,22,23]. However, as noted in previous paragraph, the association between SES and particularly FMS levels seems to be culturally dependent. While in well-developed or Western countries a positive association between SES and FMS was found from pre-school age [24,25,26], in developing or middle developed countries, including South African (SA) children, the results are not clear [27], since those with lower SES usually performed better in FMS than their peers with higher SES [28]. In addition, Armstrong [29] concluded that in SA children an inverse relation between SES and FMS was observed only for kicking the ball, which is a specific variable performed within the manipulation construct. Moreover, it is important to realize that FMS development is evidently sex-dependent during pre-school and school age. Girls usually have better locomotor skills, while boys perform better in object control [24,30,31], regardless of income status of country [32]. Pienaar et al. [5] found the same pattern in SA first grade children recruited from households with low-to-middle SES. Interestingly, some studies on pre-school [33] and school children [34] found that FMS level is rather associated with SES in girls; however, when considering age, the relation is reversed.

PF as physical readiness showed itself to be also sex-dependent when, regardless of SES, boys outperformed girls in the majority of PF capacity tests (strength, endurance, speed) from preschool age, while girls achieved better results in flexibility [35,36,37]. However, results from studies investigating the association between PF and SES are inconsistent and copy the trends of results found in the association between SES and FMS. While the majority of previous results suggest that children with higher SES tend to perform better in PF compared to peers with lower SES [29,38,39], some studies did not find any significant association [40]. Moreover, [41] or Freitas et al. [42], for instance, showed that children with higher SES performed better in speed and strength but significantly worse in flexibility and endurance compared to children from lower SES schools.

Previous findings have indicated that the currently accepted relationships between obesity, FMS and PF, for example within the Stodden model, can be strongly influenced by SES, but not always in the same direction and not similarly in both sexes. If we consider obesity prevalence, FMS and PF level from preschool age to be significantly inversely related to the amount and intensity of physical activity [43,44], then sociocultural factors, as suggested by [45], such as the opportunities for leisure time physical activities, personal family support or physical transportation habits [46], might explain why SES is differently associated with motor development or obesity prevalence in school children in culturally different countries. Therefore, the researchers deemed it essential to highlight the importance of SES in the interpretation of the links between motor development and the prevalence of obesity in children. This information might fundamentally contribute to adequate long-term motor and overall development of children.

The aim of the current study was therefore to investigate the differences in FMS, PF and body fat percentage (BF%) between Grade One learners from different ethno-geographic backgrounds in Cape Town, South Africa.

## 2. Materials and Methods

### 2.1. Participants

The measurements took place in the first term of the school year. Two Grade One classes from schools in Cape Town, South Africa, were selected to participate in the study. The two schools were based in different socio-economic areas and categorized under different quintiles. All South African schools are classified in five different quintiles based on financial resources (Table 1).

Quintile 1 schools are the poorest, while quintile 5 schools are the least poor [47]. In the current study, school B is categorized under quintile 5 and school W quintile 2. Prior to the data collection, ethical approval was obtained from the Research Ethics committee (#8456) and the study was conducted according to the guidelines of the Declaration of Helsinki. Permission was sought from the Education Department to approach the schools. Written consent from the parents/guardians and assent from the children were obtained for participation in the study. A total of *n* = 191 (*n* = 106 boys and *n* = 85 girls; 6.1 ± 0.4 years old) Grade One children participated in the study.

According to Table 2, all age categories were normally distributed. As reported by the two-way ANOVA, children from school “W” were significantly younger than children from school “B” (F _137 1_ = 8.71 *p* = 0.004; η^2^ρ = 0.07). The results also showed no significant difference between girls and boys.

### 2.2. Instruments

Anthropometric measurements:

All anthropometry parameters were measured by one trained examiner using the Eston and Reilly [48] manual. Body height was measured using a portable anthropometer P375 (Co. TRYSTOM, spol. s r.o./1993–2015 www.trystom.cz, accessed on 23 November 2021), with measurements taken to the nearest 0.1 cm. Body weight was measured using a medical calibrated scale TPLZ1T46CLNDBI300, with body weight recorded to the nearest 0.1 kg. Skinfolds were done using subscapular and triceps with the Harpenden skinfold caliper, with an accuracy of 0.2 mm. Waist circumference was measured with metal measuring tape with an accuracy of 0.1 cm.

Body fat percentage was estimated according to [49]’s equations. Previous studies showed that Slaughter’s equation is adequate alternative used in children for estimating percentage of body fat where Dual-energy X-ray absorptiometry (DXA) is not available [50,51,52,53,54].

The following equations were used [49]:

For white males with a sum of skinfolds less than 35 mm the following equation was used:BF% = 1.21 × (tric + subsc) − 0.008 × (tric + subsc)^2^ − 1.7

For black males with a sum of skinfolds less than 35 mm the following equation was used:BF% = 1.21 × (tric + subsc) − 0.008 × (tric + subsc)^2^ − 3.2

For all females with a sum of skinfolds less than 35 mm the following equation was used:BF% = 1.33 × (tric + subsc) − 0.013 × (tric + subsc)^2^ − 2.5

For males with a sum of skinfolds higher than 35 mm the following equation was used:BF% = 0.783 × (tric + subsc) + 1.6

For females with a sum of skinfolds higher than 35 mm the following equation was used:BF% = 0.546 × (tric + subsc) + 9.7

Note: tric: triceps skinfold; subsc: subscapular skinfold

The waist-to-height ratio index was used as an indirect parameter for estimating abdominal fat. Several studies have indicated that WHtR is useful in clinical and population health as it identifies children with excessive body fat [55] and greater risk of developing weight-related cardiovascular disease at an early age [56]. The waist circumference was measured at the midway between the lowest border of the rib cage and the upper iliac crest to the nearest 0.1 cm [57]. Anthropometric measurements were conducted before lunch time.

b.Fundamental movement skills:

Fundamental movement skills were evaluated with the Test for Gross Motor Development-2 (TGMD-2) [58], which is a valid and reliable measurement of FMS [59,60,61,62,63,64]. The TGMD-2 assesses proficiency in two motor-area composites (Table 3):

Inter-rater reliability for the TGMD-2 was ensured by two experienced *Kinderkineticists; both received the same videos of 10 children and had to score them according to the TGMD-2 criteria and manual.

The testing took place in the specific school’s hall. A clear demonstration of every skill was given by the assistant at the station. Children had one practice trial and two formal test trials. The formal testing trials were video recorded (consent was given) in order to properly score each participant afterwards according to the criteria of the TGMD-2 manual. The raw scores were converted to standard scores considering the sex and age of participants. Each child received a number for all the measurements and the number was shown on the video.

*Kinderkinetics is a profession that aims to develop and enhance the total well-being of children between 0–12 years of age, by stimulating, rectifying and promoting age-specific motor and physical development [65].

After the testing, the videos were transferred from the tablets to a memory stick and analysed on a computer by the researcher and assistants.

c.Physical fitness:

Physical fitness was measured using five widely accepted tests [66,67,68], namely broad jump for dynamic strength of lower limbs; throwing a tennis ball for dynamic strength of upper limb and trunk; 4×10 m shuttle running for speed agility; 20 m shuttle run endurance test (Leger test) for cardiorespiratory fitness; and a sit and reach test for flexibility. The examiners explained and demonstrated all PF tests to the children before the tests commenced. Detailed descriptions of the PF tests used are available at: (https://ftvs.cuni.cz/FTVS-726-version1-physical_fitness_tests_description.pdf, accessed on 23 November 2021).

### 2.3. Statistical Analysis

Normality of data was analysed using the Shapiro–Wilk test as well as coefficients of skewness and kurtosis. Variance–covariance homogeneity was verified using the Box M test, and the regression slopes homogeneity via the significance of between-subjects effects [69]. To accommodate age differences between children with different SES, an analysis of covariance 2 (SES) × 2 (sex) using age as covariate was applied. ANCOVA was used for selected variables, which passed all assumptions for its application (height, weight, skeletal robustness and physical fitness tests).

The effect size was estimated by the partial eta squared (η^2^ρ) with range <0.05 small effect size; 0.06–0.25 moderate effect size; 0.26–0.50 large effect size; >0.50 very large effect size [70,71] and Hedge’s g range <0.2 small effect size; 0.21–0.50 moderate effect size; 0.51–0.80 large effect size; >0.80 very large effect size. All data was analysed in NCSS2007 [72].

## 3. Results

### 3.1. Anthropometry

Since age is significantly correlated with personal height, the current study used analysis of covariance (ANCOVA) (r = 0.47), where age was determined as a covariate. Although children with lower SES were significantly younger, the difference in height in relation to SES between children with lower SES and higher SES remained significant. It means that even though the age of participants was significantly related to height (F _137 1_ = 25.63 *p* < 0.001; η^2^ρ = 0.20), children with lower SES were still significantly shorter compared to their peers with higher SES (F _137 1_ = 40.23 *p* < 0.001; η^2^ρ =0.30). Furthermore, weight was poorly correlated with age; therefore, a simple two-way ANOVA was performed, which showed that children with lower SES were significantly lighter (F _137 1_ = 39.74 *p* < 0.001; η^2^ρ = 0.30). No significant differences were found for height and weight between boys and girls. Body fat percentage and WHtR were not normally distributed (Table 4. BF% was found to be significantly greater in children with higher SES (Z = 6.04 *p* < 0.001; Hedg = 0.54). In addition, girls had a greater BF% compared to boys (Z = 4.41 *p* < 0.001; Hedg = 0.38). The same differences were found in WHtR, where children with higher SES had significantly greater values (Z = 3.89 *p* < 0.001; Hedg = 0.32). In contrast to BF%, no significant differences were found between boys and girls.

### 3.2. Fundamental Movement Skills

In general, children with lower SES achieved significantly better TGMD-2 standard scores compared to their peers with higher SES (F _137 1_ = 6.73 *p* = 0.01; η^2^ρ = 0.05). Detailed analysis, however, revealed the effect of SES only in the locomotor subtest, where children with lower SES achieved significantly better scores (F _137 1_ = 6.11 *p* = 0.014; η^2^ρ = 0.05). In the object control subtest, no significant difference was found between children with higher and lower SES; however, ANCOVA showed that girls performed better than boys (F _137 1_ = 20.78 *p* < 0.001; η^2^ρ = 0.16) (Table 5).

### 3.3. Physical Fitness

Not all results from the physical fitness tests were significantly related to SES. The effect of SES on muscular strength of trunk and upper limb—throwing (right) (F _137 1_ = 24.64 *p* < 0.001; η^2^ρ = 0.18), throwing (left) (F _137 1_ = 4.68 *p* = 0.03; η^2^ρ = 0.04) and flexibility (F _137 1_ = 12.37 *p* < 0.001; η^2^ρ = 0.09)—was found mainly in children with lower SES. In dynamic strength of lower limbs (broad jump) and agility (shuttle run—4 × 10 m), sex was found to have an effect, but not SES. In both tests, girls achieved lower dynamic strength of lower limbs (F _137 1_ = 10.04 *p* = 0.002; η^2^ρ = 0.08) and were significantly slower compared to boys (F _137 1_ = 8.16 *p* = 0.005; η^2^ρ = 0.06). No significant difference was found in cardiorespiratory performance (CRP) (Beep test), even though children with lower SES achieved better results (Table 6).

## 4. Discussion

The aim of the current study was to investigate the differences in FMS, PF and BF% between Grade One learners from different socio-economic backgrounds in Cape Town, South Africa. After controlling for differences in sex and age, SES was positively associated with height and weight.

Children with higher SES had significantly higher BF% and were heavier. Similar results brought [73] to the conclusion that overweight and obese children in China are from a higher SES. Possible reasons include available amounts of food, less physical activity and a more sedentary lifestyle in children with higher SES. These findings are contrary to the results of [74,75], who found that weight and body mass index in relation to obesity of British children with lower SES were higher compared to their peers with higher SES. In addition, higher weight and body fat are considered as a sign of wealth in certain countries [76,77]. For instance, children with high SES in Sub-Saharan Africa also displayed a higher chance of being overweight and obese [78]. These findings contradict previous studies [9,12,13] which found an inverse association between SES and body fat. A multiethnic study [79] found that obese African black girls had the highest self-esteem compared to Asian or European peers. Specifically, overweight South African black women perceive themselves as more attractive [80,81]. A very recent qualitative study [82] revealed in South African adult participants that fatness is connected with symbols of prosperity and beauty rather than with health problems. A different view of body status is also known from other cultures such as China, where being too thin is the same problem as being too fat [83]. This suggests that socio-cultural environments including ethnicity or race can link SES to weight gain and obesity status differently, as proposed by [84].

### 4.1. Fundamental Movement Skills

The results of this study indicate that children with lower SES performed significantly better than their higher-SES peers according to the standard scores of total TGMD-2 and the locomotor subtest of the TGMD-2, but not in the object control subtest. This finding is in contrast with most previous research from the Western world, where children with higher SES outperformed their lower-SES counterparts in FMS [85,86]. For instance, [34], who performed their study in Australia, and [86], who performed theirs in the United States, suggested that the differences could be attributed to lower cardiorespiratory fitness, physical activity levels, absence of weekly physical education, fewer opportunities for perceptual motor experiences and disadvantaged communities that lack facilities. Nevertheless, our study found no significant difference in CRF in relation to SES (see in detail below PF part). Furthermore, our findings are consistent with a recent South African study by [87], which was carried out in a very similar demographic environment and which also stated that rural low-income children had significantly better TGMD-2 standard scores. This negates the notion that children with lower SES naturally perform worse in overall FMS than children with higher SES due to limited access to safe outdoor playing and equipment [88] or safe places to be active in the community or to sporting equipment at home [89]. A possible reason for this finding is that children with lower SES often engage in unstructured moderate-to-vigorous physical activity with limited teacher facilitation compared to their higher-SES peers, and this might positively influence the development of FMS [87,90,91]. Children with lower SES in South Africa also tend to spend a greater amount of time in active transportation to and from schools [92,93]. Therefore, according to some authors, different findings in terms of FMS levels in South Africa compared to western, educated, industrialized, rich and democratic (WEIRD) countries can be attributed to South Africa’s unique socio-cultural environment [94,95].

### 4.2. Physical Fitness

In the physical fitness measurement, only the performances in upper limb throwing and flexibility were significantly inversely associated with SES, where children with lower SES achieved significantly better results. These findings support those of [96], who suggested that the relationship between PF and SES has not been consistently clarified.

Some studies, however, did observe a significant positive link between SES and aspects of PF (muscle strength, aerobic fitness, muscular endurance and speed) [97,98,99]. In contrast, [99] did not find any provable associations between SES and PF, and other studies [39,100] even found that children with lower SES outperformed their peers with higher SES in flexibility and endurance. The results of the current study are more consistent with the conclusions of the latter.

Inverse associations between SES and throwing could be explained by differences in opportunities and content of physical activity [29]. It has been known for more than 70 years that the way children with different SES spend their leisure time depends on their SES environment [101,102,103]. Children with higher SES usually spend leisure time participating in organized commercial physical activities [104], while children with lower SES tend to play simple group games with cheap equipment in the street [105,106]. In addition, this spontaneous type of PA usually has implicit motor learning characteristics [107], where a high number of repetitions of motor activity without explicit instructions is considered typical. Implicit motor learning for acquiring motor skills such as overhead throwing has been shown to significantly influence automation and accuracy of the movement pattern [108,109,110]. Therefore, the range of the movement experience and the defined motor pattern, along with how this motor pattern (overhead throwing) was acquired in low SES children, could explain the inverse association between SES and performance in throwing a tennis ball found in the current study. This assumption would also support previous suggestions that children with lower SES seem to have better coordination [97,111,112,113]. On the other hand, the results of the current study do not support the South African study conducted by [29], who did not find any significant differences in the throwing of a cricket ball in 6–7-year-old children when taking SES into consideration. That study included more than 600 children from five provinces in that age category. Since the participants of the current study were only from one province, sample variability could be a reason for the discrepancy in the results.

In the flexibility measurement, children with lower SES showed better results. These are in line with the findings of [29,39]. This difference could be explained by genetic differences in collagen alleles associated with physical performance/functional tests [114], even though relationships between genotypes and clinical phenotypes are not well defined. Chan et al. [115] found that in African Americans, collagen development COL1A1 COL1A2 responsible for development of bone, cartilage and tendons seemed to be evolutionarily different from European Americans, increasing flexibility in the African American population.

If one looks at differences in performance of each component of PF, it is evident in the research that children with higher SES are stronger and have better muscular explosiveness [97,111] which is in line with the current study’s findings.

Results from the cardiorespiratory fitness (CRF) test showed that children with lower SES performed better in the multistage fitness test. However, due to the large variability of results in each category of children (higher SES, lower SES, boys and girls), the differences were not significant. Nevertheless, studies from Sub-Saharan Africa found better CRF in children with lower SES compared to their higher-SES counterparts [42,105,116]. The better CRF of children with lower SES could be explained by their daily habits and physical activity profile compared to children with higher SES. Prista et al. [117] found that increased physical activity of children with lower SES was mainly due to higher demands of daily physical activities, such as walking, running and playing. VandenDriessche et al. [96] and Micklesfield et al. [103] found that children with lower SES walked to school and engaged in more physical activity on the way to and from school. On the other hand, these children spend less time in moderate-to-vigorous physical activity at school and in clubs [105]. Furthermore, Micklesfield et al. [103] suggested that children with lower SES spend more active time at the household and community level, which implies less sedentary behaviour in this social environment. These findings seem to be dependent on the social and cultural environment because the result of better PF in children with lower SES is contrary to results from studies done in the Western world, where children with lower SES performed repeatedly worse in CRF tests [99,112].

In summary, our findings should be used for the development of further education strategies with the aim of preventing obesity and properly controlling child’s motor development and physical fitness, which are influenced by SES differently considering specifics of socio-cultural and ethno-graphic experiences. It means, for instance, the extension of PE classes at least in primary school education, along with changes in content or implementation of active breaks or socialization games. This seems, according to [118], to be positively influenced by the conjunction of school and family environments intervention programs.

### 4.3. Strength and Limitations

To the best of our knowledge, this is the first study of its kind to consider traditional and cultural experiences (including ethnographic differences) as an important factor influencing the direction of the links between SES, motor performance and body fat percentage in children. An additional strength of this study is that the sample (two different socio-economic/ethno-graphic groups) was specifically defined and selected according to the guidelines (www.education.gov.za, accessed on 16 December 2021) stipulated by the Department of Basic Education in South Africa. However, the research sample was selected from a narrow population in the Western Cape; therefore, the results of this study may not be completely representative of all Grade One children in the Western Cape. Furthermore, the absence of biological maturation status of the children in this study might be a limitation because previous studies have suggested that biological maturation influences performance in strength and endurance [119,120]. However, most previous studies did not consider this parameter in similar age samples. In the current study children with higher SES were significantly taller and heavier, which suggests that they might be advanced in their biological maturation. Unfortunately, there is no valid and reliable method in South Africa to assess biological maturation for this age group in multi-ethnic populations. We suggest that future research explore the inclusion of biological maturation when assessing motor performance and BF% in children with different SES.

## 5. Conclusions

In contrast to Western countries, children with lower SES in the current study were leaner, had lower BF% and performed significantly better in FMS (specifically in their locomotor skills) compared to their higher SES peers. Furthermore, children with lower SES performed significantly better in dynamic strength of the trunk and upper limb and flexibility compared to children with higher SES. Therefore, we suggest that links between SES, PF, FMS and BF% in children seem to be dependent on country-specific cultural and ethno-graphic experiences. The uniqueness of cultural experiences with regard to SES should be included as an important factor for the development of programmes and interventions to enhance lifelong motor behaviour and health strategies for children.

## Figures and Tables

**Table 1 ijerph-19-00121-t001:** Background to South African quintile system.

Quintile	Description
1–3	No fees schools
4–5	Fee paying schools

**Table 2 ijerph-19-00121-t002:** Descriptive Age—School “W”—lower SES and School “B”—higher SES.

School	Boys Mean	Girls Mean
School “W” lower SES	6.6 ± 0.3	6.54 ± 0.33
School “B” higher SES	6.75 ± 0.39	6.73 ± 0.29

Data are presented as mean ± SD.

**Table 3 ijerph-19-00121-t003:** TGMD-2 composites.

Locomotor	Object Control
Run	Striking a stationary ball
Hop	Stationary dribble
Horizontal jump	Catch
Leap	Overhand throw
Gallop	Kick
Slide	Underhand roll

**Table 4 ijerph-19-00121-t004:** Descriptive height and weight frame indices.

	SES			
Variables	Boys (Lower SES)	Boys (Higher SES)	Girls (Lower SES)	Girls (Higher SES)
**Height** cm	116.6 ± 4.13 *** (a)	122 ± 3.98	115.3 ± 5.97 *** (b)	122 ± 5.37
**Weight** kg	20.4 ± 3.3 *** (a)	24.2 ± 3.12	19.7 ± 2.8 *** (b)	26.1 ± 7.7
**BF%**	10.6 ± 3.8 *** (a)	16.4 ± 4.9	14.6 ± 3.4 *** (b)	20.9 ± 8.1
**WHtR**	0.43 ± 0.03 *** (a)	0.45 ± 0.03	0.44 ± 0.03 *** (b)	0.47 ± 0.05

Data are presented as mean ± SD, *** *p* < 0.001. (a) Significant difference between boys with lower and boys with higher SES. (b) Significant difference between girls with lower and girls with higher SES.

**Table 5 ijerph-19-00121-t005:** TGMD-2 performance considering SES and sex of children.

TGMD Skills	Boys (Lower SES)	Boys (Higher SES)	Girls (Lower SES)	Girls (Higher SES)
Object control	10.2 ± 1.5	10.1 ± 2.1	11.8 ± 2.2 *** (b)	11.6 ± 2.0 ***
Locomotor	10.1 ± 3.0 * (a)	8.3 ± 1.8	9.6 ± 2.2 * (b)	8.8 ± 2.3
Overall TGMD-2	20.3 ± 3.0 ** (a)	18.4 ± 3.2	21.4 ± 3.0 ** (b)	20.4 ± 3.2

Data are presented as mean ± SD * *p* < 0.05 ** *p* < 0.01 *** *p* < 0.001. (a) Significant difference between boys with lower and boys with higher SES. (b) Significant difference between girls with lower and girls with higher SES.

**Table 6 ijerph-19-00121-t006:** Physical fitness performance considering SES and sex of children.

	SES			
	Boys		Girls	
	(Lower SES)	(Higher SES)	(Lower SES)	(Higher SES)
Broad jump (cm)	113.0 ± 16 ** (c)	120.6 ± 17.3 ** (c)	105.2 ± 20.2	108.0 ± 19.9
Shuttle run—4 × 10 m (s)	13.8 ± 1.1 ** (c)	13.7 ± 1.3 ** (c)	14.3 ± 0.8	14.4 ± 1.3
Throw right (m)	15.1 ± 5.5 *** (a)	10.7 ± 3.7	10.0 ± 4.0 *** (b)	7.6 ± 1.6
Throw left (m)	7.2 ± 3.0	6.5 ± 2.4	6.7 ± 2.8	5.9 ± 1.5
Beep test (No. of tracks)	16.8 ± 9.6	15.2 ± 6.9	15.8 ± 6.7	13.6 ± 6.7
Flexibility (cm)	19.5 ± 4.7 ***	15.6 ± 4.9	20.7 ± 5.0 ***	18.4 ± 5.9

Data are presented as mean ± SD ** *p* < 0.01 *** *p* < 0.001. (a): Significant difference between boys with lower and boys with higher SES. (b): Significant difference between girls with lower and girls with higher SES. (c) Significant difference considering sex regardless of SES.

## Data Availability

The data set is available at https://www.researchgate.net/publications/create?publicationType=dataset, accessed on 17 December 2021.

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
