# Peer review of "The Influence of Cultural Experiences on the Associations between Socio-Economic Status and Motor Performance as Well as Body Fat Percentage of Grade One Learners in Cape Town, South Africa"

_ijerph, 2021, doi:10.3390/ijerph19010121_

Round 1
Reviewer 1 Report
The main aim of this study was to investigate links between SES and FMS, PF, BF% of Grade one learners from two different areas in Cape Town. Regarding the authors, I would like to congratulate and thank them for their effort and motivation involved in this research study. The presentation of the research is well documented, with a scientific basis and respects the latest standards regarding the highest level scientific publications. The conclusions support and result from the research carried out and open new directions for future research.
It is probably the first study of its kind to consider traditional and cultural experiences as an important factor influencing the direction of the links between SES, body fat percentage and motor performance in children. The methodology was chosen correctly. I am curious to see what data an extended study with quintile 1, 3 and 4 groups would provide, but of course it is also valid in its current form. The submitted work is interesting and essentially exhausts the subject under discussion.
The only thing I would like to point out is that the entire manuscript needs to be brought into compliance with the requirements outlined in the MDPI Instructions for Authors section. Many of the footnotes (for example, those referring to the citation of articles from scientific journals) were improperly edited and need to be corrected and made consistent. However, this is only a technical thing that we can qualify as a need for minor corrections. Authors must also add Author Contributions, Institutional Review Board Statement, Informed Consent Statement and Data Availability Statement. I consider the latter to be particularly important, as the reader may wish to see detailed and statistically unprocessed study data, which perhaps the authors could add as supplementary material to the article.
In summary: I consider the research carried out to be novel and extremely important, which should necessarily be published as a research article. With the above mostly technical additions, the article will be suitable for publication in the International Journal of Environmental Research and Public Health.
Author Response
Dear reviewer,
we appreciate your interest about our work and also comments you did.
Author contribution is beyond of acknowledgement session. We added information abotu ethic approval, inform consent information int section 2.1 participants. However do you think about some specific information we have to add?
We alo extended Introduction section as well as Discussion part in sense to better point on assocations between SES and motor as well as somatic development in different cultures
We also tried to better format footnotes
Reviewer 2 Report
please, improve some part of the paper.
SA was not specified in introduction
2.1 use a table to summarize the quintiles os schools. this solution can improve the readability
the number of protocol is due for Helsinky declaration
2.2. erase the web address
add a reference about the measure of waist circumference
in the tables please add specification about the differences. are that about counterparts?
Author Response
First of all we hjughly appreciate your effort!
SA was not specified in introduction - corrected
2.1 use a table to summarize the quintiles os schools. this solution can improve the readability - corrected table was added
the number of protocol is due for Helsinky declaration - corrected
2.2. erase the web address - we are not sure that deleting web address that links to detial description of physical fitness test is suitable
add a reference about the measure of waist circumference - added
in the tables please add specification about the differences. are that about counterparts? corrected
Reviewer 3 Report
Thank you for the opportunity to review this article. All my comments and suggestions are aimed at improving the quality of the paper.
This may become an interesting study with some enhancement in specific sections of the article. Beneath there some comments and suggestions.
Abstract - I my opinion the abstract although it is informative seems too long - there are too many details which are not needed in that abstract - like details concerning measuring tests.
Key words are ok.
Introduction is too concise, too short so to say, to deal with such a broad topic - I think since you have some variables that you are going to analyze in the study those should be introduced in the Introduction section- what is the state of body mass issue in children of the age-category you are going to present in your study (look for same papers on on assessment of epidemiological obesity), albo gross motor skills (maybe you should look to gender/sex differences between boys and girls to see = Are there any differences between first grade boys and girls? in physical fitness or BMI?)
Also the issue of parents educational level or socio-economic status should be introduced in that part of the text (maybe look into: Social class, parental education and obesity prevalence in a study of six-year old children in Germany), especially that some of the health/obesity-related problems in children are derived from the problems in their relations with parents (Parents's social status and children's daily physical activity: The role of familial socialization and support or Health-related fitness components-links between parents and their child).
Strengthening the rationale for the study in my opinion is a must.
Methods - the use of research tools and methodology is clear and rises no concerns. It is well-described and sufficient.
Results - the table 2 concerning descriptive characteristics of the sample in my opinion should be moved to the Methods section as subsection - Participants, as it is not the results of your study but description of the group.
The rest of results is presented in a clear manner and tables are neat.
Discussion - this section would also require some more work, especially I would expect more links to culture impact on the PA and BMI - for example how body image of women and men are presented in the local media? what is the ideal of female body in terms to culture, tradition or religious perspectives. Especially that you refer to the country and ethno-graphic context. Also, I think it would wise to present findings from some studies dealing with such problems that proved positive effects in other regions or countries (like in terms of obesity problems: Effects of PA intervention on body composition in young children: influence of body mass index status and gender, or Does school-based PA decrease overweight and obesity in children ages 6-9 years? A two-year non-randomized longitudinal intervention study or in terms of gross motor skills: BRAINballs program improves the Gross Motor skills of Primary School Pupils in Vietnam) to indicate solutions and recommendations instead of just presenting the state of art.
Author Response
Dear reviewer,
we highly appreciate your effort and suggestions.
In Introduction we extent parts related to obesity prevalence, associations between obesity nad SES, FMS and SES either PF and SES
Further, according other reviewer comment we also work and corrected Method parts with including ncessayr details
We also added important information to Discusssion section. Specifically we extent the part how cultural differences in percieving of fatness, obesity or body image status might differently influence reciprocal associations between PF obesity, and FMS. We also did short summarization in the end of discussion part emhasizing the suggestion how our findings could be used practically.
On the other hand we deeply though about implemetnation of findings related to efficacy of various movement or physical programes either in leisure time or school based time. However, due to main aim fo this study we concluded it could more confusing than contributing for reader, we want to keep clear line of presented research topic.